# MicroRNA166: Old Players and New Insights into Crop Agronomic Traits Improvement

**DOI:** 10.3390/genes15070944

**Published:** 2024-07-18

**Authors:** Zhanhui Zhang, Tianxiao Yang, Na Li, Guiliang Tang, Jihua Tang

**Affiliations:** 1National Key Laboratory of Wheat and Maize Crop Science/Collaborative Innovation Center of Henan Grain Crops/College of Agronomy, Henan Agricultural University, Zhengzhou 450002, China; jiangxiaoyu0604@163.com; 2Plant Molecular and Cellular Biology Program, University of Florida, Gainesville, FL 32611, USA; tianxiao.yang@ufl.edu; 3Department of Biological Sciences, Michigan Technological University, Houghton, MI 49931, USA; gtang1@mtu.edu; 4The Shennong Laboratory, Zhengzhou 450002, China

**Keywords:** microRNA166, *HD-ZIP III genes*, plant development, stress response, agronomic traits improvement

## Abstract

MicroRNA (miRNA), a type of non-coding RNA, is crucial for controlling gene expression. Among the various miRNA families, miR166 stands out as a highly conserved group found in both model and crop plants. It plays a key role in regulating a wide range of developmental and environmental responses. In this review, we explore the diverse sequences of *MIR166s* in major crops and discuss the important regulatory functions of miR166 in plant growth and stress responses. Additionally, we summarize how miR166 interacts with other miRNAs and highlight the potential for enhancing agronomic traits by manipulating the expression of miR166 and its targeted *HD-ZIP III* genes.

## 1. Introduction

Due to the ongoing impact of climate change, crop production is facing significant challenges from extreme temperatures, drought, and flooding [1,2]. It is crucial to optimize agronomic traits and develop more resistant varieties. Therefore, exploring novel regulatory players and their biological functions is required for crop enhancement [3,4,5]. Despite engineering protein-encoding genes, manipulating miRNAs and their targets also provides a promising method for crop improvement. miRNAs are small, single-stranded, non-coding RNA molecules that play a critical role in post-transcriptional gene regulation in plants. *MIRNA* genes are transcribed and cleaved into a miRNA duplex by Dicer-like 1 (DCL1) and other D-body-related proteins. miRNA duplexes are then recruited by the Argonaute1 (AGO1) protein and incorporated into RNA-induced silencing complexes (RISCs) [6]. The miRNA-RISCs negatively regulate target gene expression via mRNA cleavage within the miRNA complementary site [7,8] or by translation inhibition [9,10]. miRNAs are essential for controlling a wide range of developmental and environmental processes by targeting specific transcription factors at the post-transcriptional level [11]. Here, we focus on miR166, a miRNA known to modulate complex agronomic traits and responses to abiotic stress in major crop species [12,13]. In this study, we discuss the sequence diversity of *MIR166* in different plant species and highlight the regulatory role of miR166 in both model and crop plants, as well as its interactions with other miRNAs. Moreover, we highlight the agronomic trait improvement by manipulating the expression of miR166 and its targets, *Class III HD-ZIP transcription factor genes* (*HD-ZIP IIIs*).

## 2. Conservation and Diversification of *MIR165/166* in Model Plants and Main Crops

The miR165/166 family is both highly conserved and abundant in land plants [14,15]. miR166 genes have been identified in land plants, while miR165 genes have only been identified in the *Brassicaceae* family [16,17,18]. To explore the sequence diversity of miR165/166 in land plants, mature miRNA sequences from six dicots and four monocots were obtained and aligned manually using Clustal omega software (Release 22.1) [18]. Among the dicots, *Arabidopsis* (*Arabidopsis thaliana*), rapeseed (*Brassica napus*), soybean (*Glycine max*), cotton (*Gossypium hirsutum*), alfalfa (*Medicago truncatula*), and tomato (*Solanum lycopersicum*) contain 9, 6, 21, 2, 7, and 3 miR165/166 members, respectively (Figure 1). In the monocots, stiff brome (*Brachypodium distachyon*), rice (*Oryza sativa*), sorghum (*Sorghum bicolor*), and maize (*Zea mays*) have 10, 14, 13, 11, and 14 miR166s, respectively (Figure 1). Mature miR165/166 sequences are highly conserved, reflecting their similar functions within these species (Table 1). The *Arabidopsis* genome has two miR165s (miR165a and miR165b) and seven miR166s (miR166a-miR166g). miR165 and miR166 have almost identical nucleotide sequences except for a C-U substitution at the 17th base, which has been confirmed with their distinct action mechanisms [14]. Similarly, there are minimal nucleotide variations in members of the miR165/166 family from rapeseed, soybean, cotton, alfalfa, and tomato. Monocots exhibit a larger number of members and diverse nucleotides in the miR166 family as compared to dicots. Notably, the maize miR166 family displays 4 different nucleotides, while miR166g in stiff brome shares only 11 conserved nucleotides with other family members. Overall, monocots likely have more target genes regulated by the miR165/166 family than dicots.

To analyze the evolution of *MIR165/166s*, a phylogenic tree was constructed using the hairpin sequences of miR165/166 from various species, including *Arabidopsis*, rapeseed, soybean, cotton, alfalfa, tomato, stiff brome, rice, sorghum, and maize (miRbase release 22.1) [19]. A total of 96 *MIR165/166s* were obtained and further classified into 7 clades, with 2 dicot-specific clades (consisting of 19 and 14 *MIR165/166s*, respectively), 2 monocot-specific clades (including 7 and 10 *MIR166s*, respectively), and 3 mixed clades (11, 23, and 12 *MIR166s*, respectively). All *MIR165/166s* in *Arabidopsis*, rapeseed, and soybean can be grouped to dicot- specific clades, and most *MIR166s* are not specific to monocot species. *MIR166s* in each monocot species were grouped into five clades (two monocot-specific clades and three mixed clades), indicating a greater diversity of *MIR166s* in monocots as compared to dicots. 

In eukaryotes, *MIRs* primarily originate from inverted duplications, random hairpin sequences, and small transposable elements [7,20,21]. Tandem and segmental duplications in plant genomes contribute to the diversification of *MIRs* [22]. Several miRNA clusters have been found in plants. For instance, miR166s can be transcribed from a single polycistronic transcript [23]. In the six dicots and four monocots mentioned above, polycistronic *MIRs* exist in rapeseed, soybean, cotton, alfalfa, rice, maize, stiff brome, and sorghum (Table 2), and represented by *bna-MIR166b-c*, *gma-MIR166e-q*, *osa-MIR166i-j*, *osa-MIR166h-k*, *zma-MIR166k-m*, *bdi-MIR166h-j*, and *sbi-MIR166f-g*. Additionally, *bna-MIR166a-e* have two copies in the soybean genome. 

## 3. Functions of miR166 in Crop Development and Stress Response

### 3.1. miR166 as a Determinant in Plant Morphogenesis

In land plants, miR165/166 is a crucial regulator in leaf polarity establishment, shoot meristem formation, and ovule and floral development (Figure 2) [18,24,25,26,27,28,29,30,31,32,33]. In *Arabidopsis*, mutants involving miR165/166 and its targets exhibit aberrant leaf polarity [34,35]. Specifically, miR165a, miR166a, and miR166b are expressed on the abaxial surface, while *PHABULOSA* (*PHB*) and *REVOLUTA* (*PHV*) are expressed on the adaxial surface, contributing to the establishment and maintenance of leaf polarity. The role of miR165/166 in leaf polarity regulation has been demonstrated in other dicot crops, such as cotton, tomato, and tobacco [12,27,36,37]. In monocot crops like rice, maize, and wheat, miR166 performs similar functions [13,38,39,40,41]. The knockdown of rice miR166 mediates leaf rolling by releasing its targeted *homeodomain containing protein4* (*OsHB4*) mRNA [38]. In maize, the miR166-*rolled leaf 1/2* (*Rld1/2*) regulatory module interacts with the miR390-*leafbladeless1* (*lbl1*) regulatory module to define the expression of ta-siRNA, establishing concentration gradients and maintaining leaf polarity [42,43,44]. In wheat, the loss control of *HB2* from miR165/166 also mediates rolled leaf [41]. 

The shoot apical meristem is responsible for generating aboveground aerial organs throughout the lifespan of higher plants, involving complex molecular mechanisms [45,46]. miR165/166 has been shown to modulate shoot apical meristem formations [30,47,48,49]. In *Arabidopsis*, AGO10 competes with AGO1 to bind miR165/166, which is essential for shoot apical meristem development and maintenance [30,50]. Sequestration miR165/166 by AGO10 has also been shown to fine-tune the axillary meristem initiation [49]. In rice, several *HD-ZIP III* genes regulate leaf initiation via an auxin-dependent manner [43]. The miR166-*HD-ZIP III* module controls maize inflorescence development and defines tassel architecture through interacting with ZmAGO18b [13,51]. In both model plants and major crops, the regulation of the shoot apical and axillary meristem development by miR166 subsequently affects flowering time, plant height, and fruit size [11,12,13,38,41]. For instance, the overexpression of *RDD1*, a target gene of rice miR166 in vascular tissue, enhances nutrient absorption, transportation, assimilation, and photosynthesis, thus resulting in higher grain yield [52,53].

In addition to the impacts on leaf polarity establishment and shoot meristem formation, miR166 has also been found to regulate plant reproductive development in several plant species. In *Arabidopsis*, miR165/166 is highly expressed in ovule primordia, which restricts the *PHB* expression and promotes integument formation, thereby influencing ovule morphogenesis [18]. In tomato, miR166 has been indicated to regulate ovule and flower morphogenesis, as well as pollen viability under adverse temperatures [27,54]. In rice, the anther adaxial/abaxial polarity is fine-tuned by the miR166-*SPOROCYTELESS/NOZZLE* (*SPL*) module so as to build the internal boundary and establish the internal structure for the anthers [55]. Point mutations in the binding site between miR166 and the *HB2* gene cause abnormal spikes in wheat [41].

### 3.2. miR166 Regulates Root and Vascular Development

Roots, the underground organs of plants, provide essential functions such as water and nutrient uptake, as well as anchorage for plant survival. Root development is intricately regulated by transcription factors, miRNAs, phytohormones, and environmental cues [56,57]. An increasing number of studies have shed light on the roles of miR166 in root development (Figure 3A–C). In *Arabidopsis*, *MIR165a* and *MIR166b* are activated by transcription factors SHORT ROOT (SHR) and SCARECROW (SCR) [58]. miR165a, miR166a, and miR166b are specifically expressed in the endodermal layer, and their movements from the inner to the outer regions are crucial for vascular patterning and root architecture [58,59]. The opposing activity between miR165/166 and the *HD-ZIP III* genes coordinates root growth and development [60]. The knockdown of miR166 and the overexpression of *HD-ZIP III* gene *HB15* lead to inhibition of vascular development and secondary cell wall formation, whereas the *HB15* mutant displayed the opposite phenotype in response to high temperature [61]. In Medicago, the overexpression of miR166 leads to the reduced formation of bundles, which leads to a reduction in the symbiotic nodules and lateral roots [62]. Despite the significant differences in root systems between monocots and dicots, miR166 also influences maize root development. In maize, the interactions of miR166-*Rld1/2* and miR390-*lbl1* are involved in root development in an auxin-dependent manner [63]. Maize mutants with the inactivation of miR166 also exhibit decreased formation of lateral roots [13]. 

In addition to its role in regulating root vascular patterning, miR166 also plays a crucial role in stem vascular development (Figure 3D–E). The overexpression of *Arabidopsis* miR165/166 leads to defects in vascular tissues and interfascicular fibers [64]. In rice, miR166 is involved in xylem development, as evidenced by the aberrant vascular anatomy observed in miR166 knockdown mutants [12,38]. Furthermore, the OsmiR166b-*OsHox32* module regulates the expression of cell-wall-related genes, influencing the mechanical strength of the plants [65]. Similarly, a maize miR166 knockdown mutant shows abnormalities in stem vascular patterning [13].

### 3.3. The Regulatory Role of miR165/166 in Phytohormones Signaling

Phytohormones are signaling molecules that are involved in many developmental and environmental processes [31]. miRNAs, including miR166, serve as crucial regulators in phytohormone response pathways (Figure 4) [66]. In *Arabidopsis*, the spatiotemporal expression of miR165/166 is fine-tuned by phytohormone crosstalk [31]. Six phytohormones, including indole-3-acetic acid (IAA), gibberellic acid (GA), cytokinin (CK), abscisic acid (ABA), jasmonic acid (JA), and salicylic acid (SA) have been suggested to modulate the expression of miR165/166s, implicating their involvement in phytohormone responses. miR165/166-*HD-ZIP IIIs* modules play critical roles in *Arabidopsis* ABA homeostasis through regulating *BG1* expression [28]. In maize, the inactivation of miR166 mediates increased ABA levels and decreased IAA levels [13]. However, the ABA contents in rice miR166 knockdown mutants by short tandem target-mimic (STTM) technology are nearly unaffected [38], indicating potential differences in the miR165/166-dependent ABA regulatory pathways between maize and rice. In soybean, miR166 is essential for plant height modulation by regulating the GA level [67]. In *Arabidopsis*, a miR165/166 target gene, *PHABULOSA* (*PHB*) has been identified to activate the expression of the cytokinin biosynthesis gene [59].

### 3.4. miR166 in Response to Abiotic Stress and Pathogenic Infection

Plants are usually exposed to abiotic and biotic stresses that inhibit their growth and development. The post-transcriptional regulation mediated by miRNAs play critical roles in responding to abiotic and biotic stresses [3,4,68,69]. An increasing number of studies have highlighted the involvement of miR166 in various abiotic and biotic stress responses (Figure 5). In *Arabidopsis*, the downregulation of miR165/166 leads to the upregulation of its target gene *PHABULOSA* (*PHB*), potentially enhancing drought and cold resistance through ABA homeostasis [28], but making it sensitive to heat stress [70]. The high temperature mediates the reduced expression of *MIR166* and the elevated expression of the *HD-ZIP III* gene *HB-15* [61]. In maize, *STTM166*, the miR166 inactivation mutant, exhibits improved tolerance to drought, salinity, and high temperatures [13]. Similarly, the knockdown of rice miR166 results in enhanced drought resistance, characterized by rolled leaves and altered stem xylem architecture [38]. The miR166-*HD-ZIP III* gene module has been proven to be a crucial regulator in alfalfa (*Medicago sativa* L.) and tea plant (*Camellia sinensis*) [71,72]. Therefore, the lower expression of miR165/166 is crucial for resistance to abiotic stresses, although the underlying mechanisms may vary among plant species. In contrast, miR166 has distinct effects on pathogen infection and heavy metal stress. In rice, miR166k-166h enhances immunity by the post-transcriptional regulation of *ethylene-insensitive 2* (*EIN2*) [73]. The overexpression of miR166 or knockout of *OsHB4* leads to enhanced cadmium tolerance [15]. In tomato, the overexpression of miR166 enhances late blight resistance [74]. A recent study indicated that the sly-miR166*-SlyHB* module is a susceptibility factor to Tomato leaf curl New Delhi virus (ToLCNDV) [75]. The overexpression of sly-miR166 or the gene silencing of *SlyHB* enhances the resistance to ToLCNDV. 

Moreover, extensive small RNA profiling studies have revealed the involvement of miR165/166 in various stress responses, including drought resistance in tomato [76]; cold tolerance in *Brassica napus* [77]; heat stress responses in rice, maize, and wheat [78,79,80,81]; chromium tolerance in rice [82]; and virus infection in tobacco [83].

### 3.5. Other Functions of miR166 in Crops

Small RNA sequencing studies have revealed that miR166 may play roles in phloem fiber development in flax [84]; seed development in barley, narrow-leafed lupin, and maize [85,86,87]; seed germination in barley and maize [85,88]; seed dormancy in barley [89]; and heterosis formation in Brassica napus [77]. Collectively, a wealth of literature has highlighted the crucial involvement of miR166 in diverse aspects of plant development and stress responses. However, the interactions of miR166 with other miRNAs and its functions in modulating complex agronomic traits remain largely unresolved.

## 4. The Interactions between miR166 and Other miRNAs in Model and Crop Plants

In the intricate landscape of developmental and environmental processes, miR166 interacts with other miRNAs or components of the miRNA biogenesis pathway to carry out its biological functions (Figure 6). For example, in *Arabidopsis*, shoot regeneration inhibition and leaf polarity determination are regulated by AGO10-suppressing miR165/166 [30,50,90]. The maintenance of stem cells mediated by miR165/166 is dependent on the repression of AGO1 through miR168 targeting and cleavage [91]. The establishment and maintenance of leaf polarity involve the crosstalk between the miR390-AGO7-*TAS3* and miR165/166-*HD-ZIP IIIs* modules in *Arabidopsis* and maize [92]. The interplay between miR160 and miR165/166 fine-tunes the expression of ABA and IAA-related genes, impacting leaf development, drought tolerance, and somatic embryogenesis induction in *Arabidopsis* [93,94]. In salt-stressed potato, the opposing activities of miR166 and miR159 establish an asymmetric expression pattern for basal growth [95]. In *Arabidopsis*, the miR166-*HD-ZIP IIIs* module is essential for silencing seed dormancy and maturation genes during the vegetative phase, potentially interacting with the miR156-*SPLs* module [96]. Furthermore, miR172 and miR165/166, potentially connected through the WUS transcription factor, participate in modulating the temporal program of floral stem cells in Arabidopsis [97]. These studies collectively highlight the intricate regulatory networks in which miR165/166 is embedded.

## 5. Exploring the miR166-HD-ZIP IIIs Module to Improve Complex Agronomic Traits

In crops, the miR166-*HD-ZIP IIIs* module has been demonstrated to regulate various crucial processes such as plant mechanical strength, lateral meristem formation, nodulation, nutrition uptake, abiotic stress tolerance, and pathogenic immunity [12,13,15,38,48,52,53,62,65,73,74]. Hence, the miR166-*HD-ZIP IIIs* module holds great potential as a versatile toolbox for improving agronomic traits in crops. Given that miR166 has multiple family members and target genes with distinct temporal–spatial expression patterns, it becomes essential to finely regulate the expression of specific *MIR166* or *HD-ZIPIII* genes responsible for specific agronomic traits. For instance, editing the promoter sequence of *OsHox32* can lead to the downregulation of target genes, enhancing culm mechanical strength. Similarly, editing the promoter sequence of the polycistronic miRNA gene for OsmiR166k and OsmiR166h can result in the upregulation of miRNAs, thereby boosting rice pathogenic immunity. Interestingly, a recent study revealed that exogenous miRNAs can mediate post-transcription gene silencing in plants, offering an alternative method to modulate the expression of miR166 and its target genes [98]. For instance, feeding double-strand artificial miRNA (ds-amiRNA) for *MIR166s* enhances the abiotic stress tolerance; likewise, feeding ds-miR166 improves pathogenic immunity. Furthermore, studies have revealed that plant primary miRNAs (pri-miRNAs) encode regulatory peptides, termed miRNA-encoded peptides (miPEPs), which can specifically increase the expression of their corresponding miRNAs [99,100]. The exogenous application of miPEPs specifically increases their cognate miRNA expressions. Consequently, peptides like miPEP172c and miPEP171d have been utilized for improving agronomic traits in soybean and grapevine [101,102]. In *Arabidopsis*, pri-miR165a, pri-miR166a, and pri-miR166g encode miPEPs that are used to enhance the expression of miR166a and miR166g [100]. Similarly, certain pri-miR166 in major crops may encode miPEPs that could be beneficial for crop enhancement through external application.

However, it is crucial to note that gene editing and miRNA decoy strategies often result in mutations with severe phenotypic consequences, such as dwarf stature, seed abortion, or even plant lethality, making them unsuitable for crop breeding. [12,103,104]. For example, the knockdown of miR166 in *Arabidopsis*, rice, and maize yields positive effects on abiotic stress tolerance but also causes negative effects on developmental transition, fruit size, male fertility, and plant height [11,13,38]. In crop breeding, breeders typically opt to screen for ideal genotypes or haplotypes of *MIR166s* and their target genes and further optimize agronomic traits through marker-assisted selection (MAS). The interactions of miR166 with other miRNAs or genes, e.g., miR160, provides an alternative way to mitigate the negative effects by genetic crossing [93,94]. 

## 6. Concluding Remarks

miR166 is a well-conserved miRNA family in both dicots and monocots. Given the diverse functions of miR166 and its target genes in model plants and main crops, it is promising to exploit the miR166-*HD-ZIP IIIs* module for agronomic traits improvements. However, several hurdles should be considered. First, our knowledge of the miR166-*HD-ZIP IIIs* module is limited, particularly in crops. It is necessary to explore their diversified functions in crops. Second, the temporal–spatial expressions, the developmental–environmental responses, and the miR166 and *HD-ZIP IIIs* interactions are far from uncovered. The RNA profiling allows us to analyze the expression of miR166 and *HD-ZIP IIIs* at different cellular/tissular levels, developmental stages, and environmental stimulus. Third, the interplay of miR166-*HD-ZIP IIIs* with other miRNAs and miRNA biogenesis pathway components is still largely unknown. miRNA decay technologies and miRNA inducible CRISPR systems are optimal tools for us to investigate the interactive roles of miR166 [105,106].

## Figures and Tables

**Figure 1 genes-15-00944-f001:**
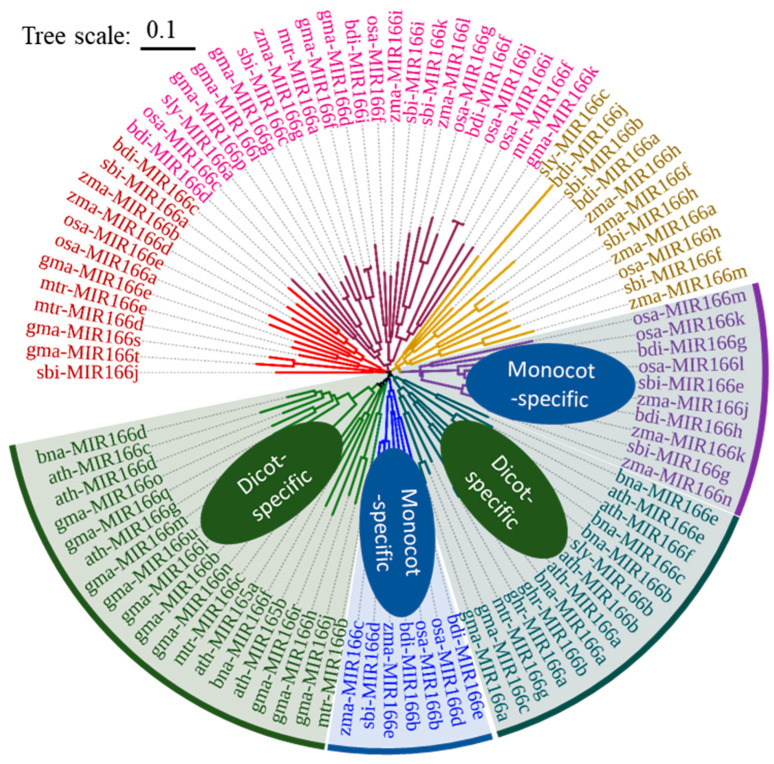
Phylogenetic analysis of *MIR165/166s* in model plants and major crops. The neighbor-joining tree was constructed using Clustal omega (V1.2.2) and iTol online software (V6). The different colors indicated the seven clades. Dicots: *arabidopsis thaliana*, *ath*; *brassica napus*, *bna*; *glycine max* (soybean), *gma*; *gossypium hirsutum* (cotton), *ghr*; *medicago truncatula* (medicago), *mtr*; *solanum lycopersicum* (tomato), *sly*. Monocots: *brachypodium distachyon*, *bdi*; *oryza sativa* (rice), *osa*; *sorghum bicolor* (sorghum), *sbi*; *zea mays* (maize), *zma*.

**Figure 2 genes-15-00944-f002:**
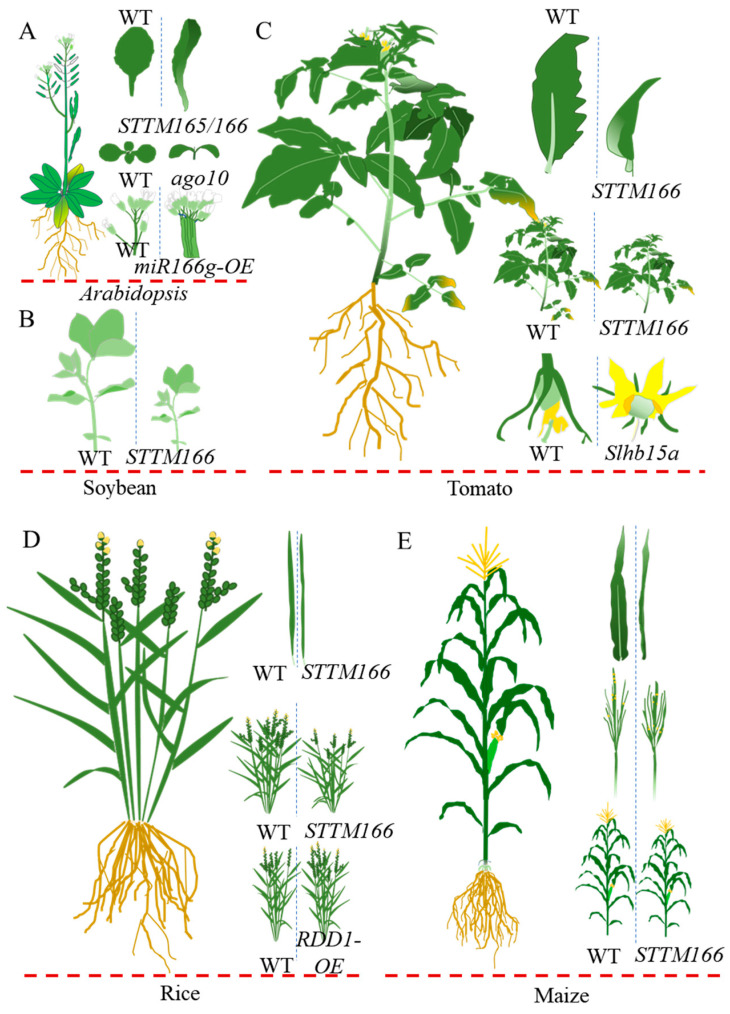
Experimentally verified functions of miR166-*HD-ZIP IIIs* in regulating model and crop plant morphology and development. (**A**). Regulatory roles of miR165/166 in *Arabidopsis* include leaf polarity, shoot apical meristem formation, and axillary meristem development. (**B**). The knockdown of miR166 leads to decreased plant height in soybean. (**C**). Tomato miR166 is involved in the regulating of leaf polarity, plant height, ovule, and flower morphogenesis. (**D**). In rice, miR166 acts as a determinant in rice leaf rolling, plant height, and yield. (**E**) The loss function of miR166 results in rolled leaf, short tassel central spike, and reduced plant height. *STTM166* represents the knockdown mutant of miR166 by short tandem target-mimic (STTM) technology.

**Figure 3 genes-15-00944-f003:**
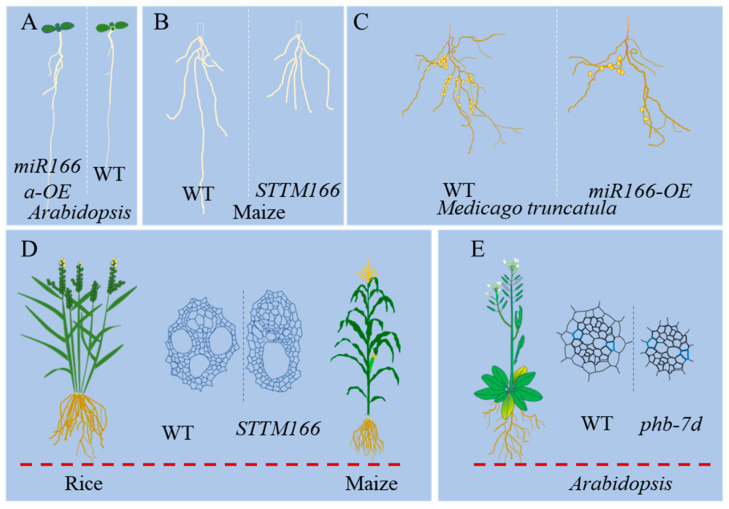
Functional identification of miR166 in model and crop plant roots and vascular development. (**A**–**C**). The overexpression or knockdown of miR166 induces root architecture alterations in *Arabidopsis*, maize, and *medcago truncatula*. (**D**,**E**). Vascular patterns determined by the miR166-*HD-ZIP IIIs* module in rice, maize, and *Arabidopsis*. *STTM166* represents the knockdown mutant of miR166 by short tandem target-mimic (STTM) technology.

**Figure 4 genes-15-00944-f004:**
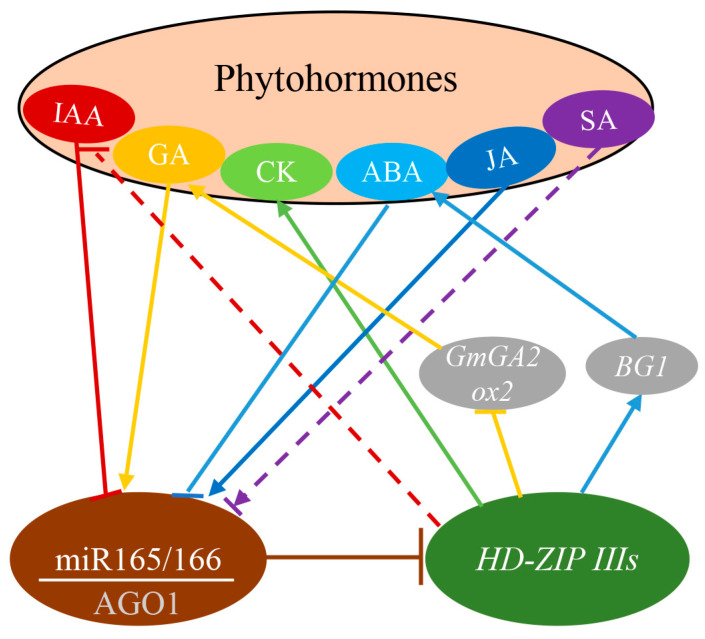
miR165/166-HD-ZIP IIIs module involved in phytochromones crosstalk.

**Figure 5 genes-15-00944-f005:**
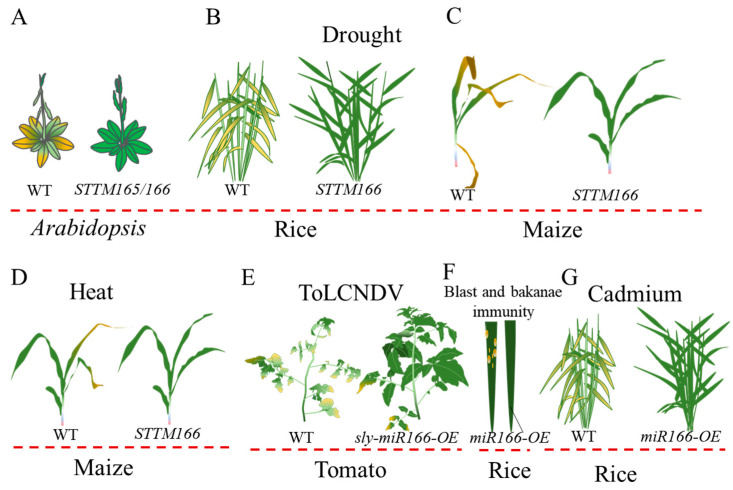
miR166 confers plant abiotic stress and pathogenic immunity. (**A**–**D**). The inactivation of miR165/166 mediates enhanced abiotic stress tolerance in *Arabidopsis*, rice, and maize. (**E**–**G**). The overexpression of miR166 is essential for improving pathogenic immunity and cadmium tolerance in rice and tomato.

**Figure 6 genes-15-00944-f006:**
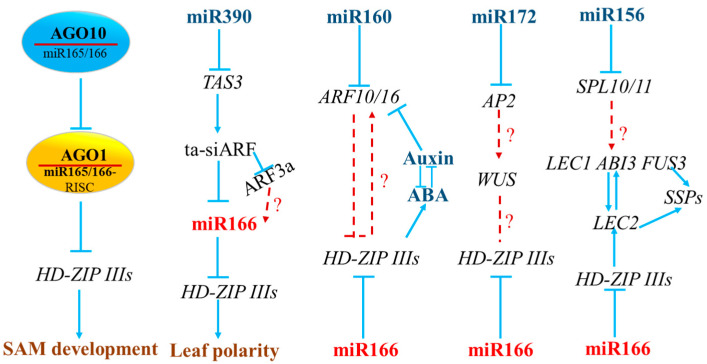
Interactive roles of miR165/166 with other miRNAs.

**Table 1 genes-15-00944-t001:** Diversification of mature miR165/166 sequences in model and main crop plants.

Species/Members	Sequence Alignment																								
* Arabidopsis thaliana *	ath-miR165a,b			U	C	G	G	A	C	C	A	G	G	C	U	U	C	A	U	C	C	C	C	C	21
9	ath-miR166a-g			U	C	G	G	A	C	C	A	G	G	C	U	U	C	A	U	U	C	C	C	C	21
* Brassica napus *	bna-miR166a-e			U	C	G	G	A	C	C	A	G	G	C	U	U	C	A	U	U	C	C	C	C	21
6	bna-miR166f			U	C	G	G	A	C	C	A	G	G	C	U	U	C	A	U	C	C	C	C	C	21
* Glycine max *	gma-miR166h,k	U	C	U	C	G	G	A	C	C	A	G	G	C	U	U	C	A	U	U	C	C			21
21	gma-miR166u	U	C	U	C	G	G	A	C	C	A	G	G	C	U	U	C	A	U	U	C				20
	gma-miR166a-g,i			U	C	G	G	A	C	C	A	G	G	C	U	U	C	A	U	U	C	C	C	C	21
	gma-miR166m				C	G	G	A	C	C	A	G	G	C	U	U	C	A	U	U	C	C	C	C	20
	gma-miR166n,o			U	C	G	G	A	C	C	A	G	G	C	U	U	C	A	U	U	C	C	C	C	21
	gma-miR166j			U	C	G	G	A	C	C	A	G	G	C	U	U	C	A	U	U	C	C	C	G	21
	gma-miR166p-t			U	C	G	G	A	C	C	A	G	G	C	U	U	C	A	U	U	C	C	C		20
* Gossypium hirsutum * 2	ghr-miR166a,b			U	C	G	G	A	C	C	A	G	G	C	U	U	C	A	U	U	C	C	C	C	21
* Medicago truncatula *	mtr-miR166a,b,d,e,g			U	C	G	G	A	C	C	A	G	G	C	U	U	C	A	U	U	C	C	C	C	21
7	mtr-miR166c,f			U	C	G	G	A	C	C	A	G	G	C	U	U	C	A	U	U	C	C	U	C	21
* Solanum lycopersicum *	sly-miR166a,b			U	C	G	G	A	C	C	A	G	G	C	U	U	C	A	U	U	C	C	C	C	21
3	sly-miR166c			U	C	G	G	A	C	C	A	G	G	C	U	U	C	A	U	U	C	C	U	C	21
* Brachypodium distachyon *	bdi-miR166g																	U	G	U	G	G	U	G	A
		U	C	U	C	G	G	A	C	C	A	G	G	C											21
10	bdi-miR166h			U	C	G	G	A	C	C	A	G	G	C	U	U	C	A	A	U	C	C	C	U	21
	bdi-miR166f	U	C	U	C	G	G	A	C	C	A	G	G	C	U	U	C	A	U	U	C	C			21
	bdi-miR166a-d,i			U	C	G	G	A	C	C	A	G	G	C	U	U	C	A	U	U	C	C	C	C	21
	bdi-miR166e		C	U	C	G	G	A	C	C	A	G	G	C	U	U	C	A	U	U	C	C	C		21
	bdi-miR166j			U	C	G	G	A	C	C	A	G	G	C	U	U	C	A	U	U	C	C	U	U	21
* Oryza sativa *	osa-miR166g-i			U	C	G	G	A	C	C	A	G	G	C	U	U	C	A	U	U	C	C	U	C	21
13	osa-miR166a-d,f,j			U	C	G	G	A	C	C	A	G	G	C	U	U	C	A	U	U	C	C	C	C	21
	osa-miR166e			U	C	G	A	A	C	C	A	G	G	C	U	U	C	A	U	U	C	C	C	C	21
	osa-miR166k-m			U	C	G	G	A	C	C	A	G	G	C	U	U	C	A	A	U	C	C	C	U	21
* Sorghum bicolor *	sbi-miR166f			U	C	G	G	A	C	C	A	G	G	C	U	U	C	A	U	U	C	C	U	C	21
11	sbi-miR166k			U	C	G	G	A	C	C	A	G	G	C	U	U	C	A	U	U	C	C	U		20
	sbi-miR166a-d,h-j			U	C	G	G	A	C	C	A	G	G	C	U	U	C	A	U	U	C	C	C		20
	sbi-miR166e,g			U	C	G	G	A	C	C	A	G	G	C	U	U	C	A	A	U	C	C	C	U	21
* Zea mays *	zma-miR166l,m			U	C	G	G	A	C	C	A	G	G	C	U	U	C	A	U	U	C	C	U	C	21
14	zma-miR166j,k,n			U	C	G	G	A	C	C	A	G	G	C	U	U	C	A	A	U	C	C	C	U	21
	zma-miR166a			U	C	G	G	A	C	C	A	G	G	C	U	U	C	A	U	U	C	C	C	C	21
	zma-miR166b-i			U	C	G	G	A	C	C	A	G	G	C	U	U	C	A	U	U	C	C	C	*	20

**Table 2 genes-15-00944-t002:** Polycistronic MIR166s in model and main crop plants.

Class	Species	Polycistronic *MIR166*	Location
Dicots	*Brassica napus*	*bna-MIR166b,c*	Scaffold2676:6222~6333 Scaffold2676:6215~6341
	*Glycine max*	*gma-MIR166e,q*	4:46797931~46798040 4:46798188~46798339
	*Gossypium hirsutum*	*ghr-MIR166a,b*	D12:41573882~41574028 D12:41573879~41574032
	*Medicago truncatula*	*mtr-MIR166c,d*	3:47901757~47901861 3:47901931~47902021
Monocots	*Brachypodium distachyon*	*bdi-MIR166h,j*	3:57184726~57184865 3:57184616~57184767
	*Oryza sativa*	*osa-MIR166i,j*	3:25294953~25295097 3:25294953~25295092
		*osa-MIR166d,h,k*	2:32435174~32435292 2:32435003~32435129
	*Sorghum bicolor*	*sbi-MIR166d,f,g*	4:64857783~64857921 4:64857514~64857647
	*Zea mays*	*zma-MIR166k,m*	5:219021288~219021455 5:219021559~219021714

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
