# Peer review of "MicroRNA166: Old Players and New Insights into Crop Agronomic Traits Improvement"

_genes, 2024, doi:10.3390/genes15070944_

Round 1

Reviewer 1 Report

Comments and Suggestions for Authors

The manuscript entitled “MicroRNA166: Old players and new insights into crop agronomic traits improvement” by Zhang et al. summarizes the conservation and diversification of microRNAs in dicots and monocots, as well as their functions in development and stress responses. This review suggests diverse sequences of MIR166s through phylogenetic analysis and comparison of mature sequences, and discusses the regulatory functions of miR166 reported in various crops, particularly in plant growth and stress responses. While this review is comprehensive, I have the following comments:

Major Comments:

1.     In your text, you mention the legend of Figure 2, but I cannot find the figure. Could you please add it correctly?

Minor Comments:

1.     Line 165: "n rice" should be corrected to "In rice".

Author Response

Comments 1: In your text, you mention the legend of Figure 2, but I cannot find the figure. Could you please add it correctly?

Response 1: We have added Figure 2 in the manuscript.

Comments 2: Line 165: "n rice" should be corrected to "In rice".

Response 2: We have revised this typing error.

Reviewer 2 Report

Comments and Suggestions for Authors

Dear authors

The manuscript provides a review of the occurrence and the biological functions of members of the miR165/166 miRNA family in various plants, including crop plant species, and their interaction with other regulatory miRNAs. The review also briefly discusses the potential to develop crops with modified traits based on a modification of expression of mi R166 levels in these crop plants. The review is quite well written and addressing an increasingly recognised topic for both plant research and biotechnology.

The manuscript, however, needs to be revised with a view to a number of issues and would benefit from further language editing. In the following general and specific comments I would like to offer suggestions for improvement.

1)    The introduction provides a short concise overview on miRNA dependent regulation. The following sections refer to miRNA regulatory modules without providing a further explanation for this term. I would suggest to explain how this term is used in the manuscript (e.g. according to the definitions which are given for example in Liu et al.; Xu et al.; Willmann und Poethig (2010; 2011; 2007)?). Furthermore, the abbreviation HD-ZIP IIIs is not introduced in a way as are e.g. DCL1 (L30) or AGO1 (L31).

2)    The color code used in Fig 1 should be indicated and explained in the caption and in text (I guess L75ff should do that but this can be improved)

3)    Fig 2 is entirely missing – pls. insert the fig!

4)    The heading of chapt 3.5. is similar to that of 3.4. which I consider is an editorial error. Pls. revise, as the chapt seems to be dealing with the function of miR166 in plant development rather than its functions in abiotic or biotic stress

5)    Pls. explain abreviated terms, such as STTM166 in Fig 3 (only referred to later in L196), ds-amiRNA and pri-miRNA in chapt 5.

Specific comments

L46: Revise or explain expression “mature sequences”

L47: Indicate used software program

L118: Reconsider “shoot apical meristem” (apical shoot meristem)

L122: Reconsider “manner” (mechanism)

L135: Clarify whether the adverse temeratures relate only to pollen or to all mentioned phenotypes

L173 Fig 4: miR165/166 should be printed in a single line; color of line from CK to HD-Zip seems to be wrong
It seems to be that GA-miR165/166-HD-Zip-GA is a positive feed back loop, while ABA- miR165/166-HD-Zip-ABA is a neg feedback loop: discuss

L273: what are miRNA decay strategies – explain or rephrase!

Liu, Bing; Liu, Lin; Tsykin, Anna; Goodall, Gregory J.; Green, Jeffrey E.; Zhu, Min et al. (2010): Identifying functional miRNA-mRNA regulatory modules with correspondence latent dirichlet allocation. In: Bioinformatics (Oxford, England) 26 (24), S. 3105–3111. DOI: 10.1093/bioinformatics/btq576.

Willmann, Matthew R.; Poethig, R. Scott (2007): Conservation and evolution of miRNA regulatory programs in plant development. In: Current Opinion in Plant Biology 10 (5), S. 503–511. DOI: 10.1016/j.pbi.2007.07.004.

Xu, Juan; Li, Chuan-Xing; Li, Yong-Sheng; Lv, Jun-Ying; Ma, Ye; Shao, Ting-Ting et al. (2011): MiRNA-miRNA synergistic network: construction via co-regulating functional modules and disease miRNA topological features. In: Nucleic acids research 39 (3), S. 825–836. DOI: 10.1093/nar/gkq832.

Comments on the Quality of English Language

L25: there is some thing missing – maybe:
“ … and develop more resistant varieties.”

L26: revise expression “underlying” (delete and or change to investigating)

L43: Consider revising to “ …is both highly conserved and abundant…”

L28-30: Move “in plants” to the end of sentence

L44: Revise expression “widely identified”

L57f: Revise to e.g. “there are minimal nucleotide variations in members of the miR165/166 family from rapeseed, soybean, cotton, alfalfa, and tomato”

L59: Revise expression “consistent” (change to conserved)

L67: Past tense seems wrong here

L103: Reconsider “conserved” (maybe: similar)

L107-108: Revise sentence

L131: Consider adding “in several plant species” at the end

L143: Revise beginning to “An increasing number of studies…”

L151-153: Revise sentence – does inducing the formation of bundles lead to a reduction in symbiotic nodules and roots?

L157: Revise to:  “exhibit decreased formation of lateral roots”

L165: Correct to “in”

L181: delete “Alternatively,”

L194: something seems to be wrong with sentence – revise

L233: Explain “autorepression”

L238: Consider changing “contrasting” to “opposing”

L255-258: Revise sentence

L260: Immunity against what?

L269ff: Somethings seems missing – maybe: “…miPEPs that are used to enhance …”

L276: Revise to “yields”

L280: Revise expression “ interactive roles”

L288: Reconsider expression “address”

L295: Revise “decipher” (investigate) and rest of sentence

Author Response

Comments 1: The introduction provides a short concise overview on miRNA dependent regulation. The following sections refer to miRNA regulatory modules without providing a further explanation for this term. I would suggest to explain how this term is used in the manuscript (e.g. according to the definitions which are given for example in Liu et al.; Xu et al.; Willmann und Poethig (2010; 2011; 2007)?). Furthermore, the abbreviation HD-ZIP IIIs is not introduced in a way as are e.g. DCL1 (L30) or AGO1 (L31).

Response 1: We have cited the related references. And the introduction of the abbreviation HD-ZIP IIIs was provided.

Comments 2: The color code used in Fig 1 should be indicated and explained in the caption and in text (I guess L75ff should do that but this can be improved)

Response 2: We have added an explanation for the color code in Figure 1.

Comments 3: Fig 2 is entirely missing – pls. insert the fig!

Response 3: We have added Figure 2 in the manuscript.

Comments 4: The heading of chapt 3.5. is similar to that of 3.4. which I consider is an editorial error. Pls. revise, as the chapt seems to be dealing with the function of miR166 in plant development rather than its functions in abiotic or biotic stress

Response 4: We have revised this editorial error.

Comments 5: Pls. explain abreviated terms, such as STTM166 in Fig 3 (only referred to later in L196), ds-amiRNA and pri-miRNA in chapt 5.

Response 5:  We have added explanations for these abbreviated terms.

Specific comments

Comments 6: L46: Revise or explain expression “mature sequences”

Response 6: We have revised “mature sequences” to “mature miRNA sequences”.

Comments 7: L47: Indicate used software program

Response 7: We have added the software information in the manuscript.

Comments 8: L118: Reconsider “shoot apical meristem” (apical shoot meristem)

Response 8: We have revised this point in the manuscript.

Comments 9: L122: Reconsider “manner” (mechanism)

Response 9: We have revised this mistake in the manuscript.

Comments 10: L135: Clarify whether the adverse temeratures relate only to pollen or to all mentioned phenotypes

Response 10: Besides of the affects on pollen viability, the adverse temperatures also relate to ovule and flower morphogenesis.

Comments 11: L173 Fig 4: miR165/166 should be printed in a single line; color of line from CK to HD-Zip seems to be wrong

It seems to be that GA-miR165/166-HD-Zip-GA is a positive feed back loop, while ABA- miR165/166-HD-Zip-ABA is a neg feedback loop: discuss

Response 14: We have checked all the related references carefully, and revised Figure 4.

Comments 12: L273: what are miRNA decay strategies – explain or rephrase!

Response 12: This is a typing error. We have revised “miRNA decay strategies” to “miRNA decoy strategies”.

Liu, Bing; Liu, Lin; Tsykin, Anna; Goodall, Gregory J.; Green, Jeffrey E.; Zhu, Min et al. (2010): Identifying functional miRNA-mRNA regulatory modules with correspondence latent dirichlet allocation. In: Bioinformatics (Oxford, England) 26 (24), S. 3105–3111. DOI: 10.1093/bioinformatics/btq576.

Willmann, Matthew R.; Poethig, R. Scott (2007): Conservation and evolution of miRNA regulatory programs in plant development. In: Current Opinion in Plant Biology 10 (5), S. 503–511. DOI: 10.1016/j.pbi.2007.07.004.

Xu, Juan; Li, Chuan-Xing; Li, Yong-Sheng; Lv, Jun-Ying; Ma, Ye; Shao, Ting-Ting et al. (2011): MiRNA-miRNA synergistic network: construction via co-regulating functional modules and disease miRNA topological features. In: Nucleic acids research 39 (3), S. 825–836. DOI: 10.1093/nar/gkq832.

Comments on the Quality of English Language

Comments 13: L25: there is some thing missing – maybe:

“ … and develop more resistant varieties.”

Response 13: We have revised this mistake.

Comments 14: L26: revise expression “underlying” (delete and or change to investigating)

Response 14: We have revised this improper expression.

Comments 15: L43: Consider revising to “ …is both highly conserved and abundant…”

Response 15: We have revised this point.

Comments 16: L28-30: Move “in plants” to the end of sentence

Response 16: We have revised this point.

Comments 17: L44: Revise expression “widely identified”

Response 17: We have this improper expression.

Comments 18: L57f: Revise to e.g. “there are minimal nucleotide variations in members of the miR165/166 family from rapeseed, soybean, cotton, alfalfa, and tomato”

Response 18: We have revised this sentence.

Comments 19: L59: Revise expression “consistent” (change to conserved)

Response 19: We have revised “consistent” to “conserved”.

Comments 20: L67: Past tense seems wrong here

Response 20: We have revised this sentence in the manuscript.

Comments 21: L103: Reconsider “conserved” (maybe: similar)

Response 21: We have revised “conserved” to “similar”.

Comments 22: L107-108: Revise sentence

Response 22: We have revised this sentence.

Comments 23: L131: Consider adding “in several plant species” at the end

Response 23: We have revised this point.

Comments 24: L143: Revise beginning to “An increasing number of studies…”

Response 24: We have revised this sentence.

Comments 25: L151-153: Revise sentence – does inducing the formation of bundles lead to a reduction in symbiotic nodules and roots?

Response 25: We have revised this sentence.

Comments 26: L157: Revise to:  “exhibit decreased formation of lateral roots”

Response 26: We have revised this improper expression.

Comments 27: L165: Correct to “in”

Response 27: We have revised this typing error.

Comments 28: L181: delete “Alternatively,”

Response 28: We have revised this point.

Comments 29: L194: something seems to be wrong with sentence – revise

Response 29: We have revised this sentence.

Comments 30: L233: Explain “autorepression”

Response 30: We have revised this improper expression.

Comments 31: L238: Consider changing “contrasting” to “opposing”

Response 31: We have revised this point.

Comments 32: L255-258: Revise sentence

Response 32: We have revised this sentence.

Comments 33: L260: Immunity against what?

Response 33: We have revised this point in the manuscript.

Comments 34: L269ff: Somethings seems missing – maybe: “…miPEPs that are used to enhance …”

Response 34: We have revised this mistake.

Comments 35: L276: Revise to “yields”

Response 35: We have revised this grammatical error.

Comments 36: L280: Revise expression “ interactive roles”

Response 36: We have revised this improper expression.

Comments 37: L288: Reconsider expression “address”

 Response 37: We have revised the improper expression.

Comments 38: L295: Revise “decipher” (investigate) and rest of sentence

Response 38: We have revised this point in the manuscript.

Round 2

Reviewer 2 Report

Comments and Suggestions for Authors

Kind thanks to the authors for diligently addressing my many comments!

Apart from a final check on grammar (at some points singular vs. plural needs to be corrected) I support the publication of the revised manuscript.

One addition may be added during final production for publication in the Caption of Fig. 2 E: all other parts (A-D) indicate the crop species which is addressed - I would suggest to include mentioning that in E this concerns maize. 

Comments on the Quality of English Language

Apart from a final check on grammar (at some points singular vs. plural needs to be corrected) I support the publication of the revised manuscript.